# FACTORED WORLD MODELS FOR ZERO-SHOT GENERALIZATION IN ROBOTIC MANIPULATION

## ABSTRACT

World models for environments with many objects face a combinatorial explosion of states: as the number of objects increases, the number of possible arrangements grows exponentially. In this paper, we learn to generalize over robotic pick-and-place tasks using object-factored world models, which combat the combinatorial explosion by ensuring that predictions are equivariant to permutations of objects. We build on one such model, C-SWM, which we extend to overcome the assumption that each action is associated with one object. To do so, we introduce an action attention module to determine which objects are likely to be affected by an action. The attention module is used in conjunction with a residual graph neural network block that receives action information at multiple levels. Based on RGB images and parameterized motion primitives, our model can accurately predict the dynamics of a robot building structures from blocks of various shapes. Our model generalizes over training structures built in different positions. Moreover crucially, the learned model can make predictions about tasks not represented in training data. That is, we demonstrate successful zero-shot generalization to novel tasks, with only a minor decrease in model performance. Furthermore, we evaluate our model on a dataset collected on a physical robot.

## 1 INTRODUCTION

From assembly to household robots, current state-of-the-art robot learning agents cannot generalize beyond a specific training task. One important aspect of generalization is the ability to understand any novel combination of known factors, a so-called combinatorial or compositional generalization. Applied to objects, combinatorial generalization ideally allows an agent to understand any arrangement of objects from only a limited number of interactions with its environment. The two key steps such agents need to perform are (a) decomposing a scene into individual objects and (b) modeling relative interactions between objects. The latter is particularly important when learning to predict the dynamics of the environment (i.e. learning a world model).

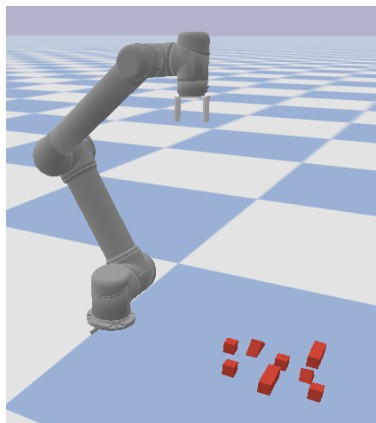

Figure 1: PyBullet environment with a UR5 arm.

The purpose of our paper is to learn a world model that accurately predicts the effects of actions in the context of robotic manipulation, and generalizes to novel tasks. We assume the state of the environment is already factored into individual images for each object, and focus on learning a latent code for each object and predicting the effect of pick-and-place actions. Our model is trained using a self-supervised contrastive loss. It can predict accurate manipulation physics by using multiple graph neural network layers (GNNs) and action attention. GNNs consider pairwise interactions, usually between every possible pair of objects, in order to predict the state of each object one step into the future. GNNs achieve combinatorial generalization (to the extent that pairwise interactions are expressive enough) by equivariance to the order in which objects are presented–permutation

equivariance. That is, they cannot overfit to a particular ordering of objects. It is a common choice to use a single GNN layer (in some cases called an Interaction Network (Battaglia et al., 2016)). But, as shown in previous work (Kipf et al., 2018; Sanchez-Gonzalez et al., 2018) and confirmed by our experiments, stacking multiple GNN layers leads to much more accurate physics predictions.

The ability to integrate information about actions with latent states of individual objects has been underexplored in prior work. Previous object-factored world models either assume object-action association (a factored action space) (Veerapaneni et al., 2019; Kipf et al., 2020; Huang et al., 2020) or only model sequences of states without actions (Janner et al., 2019; Bakhtin et al., 2019; Qi et al., 2021). We represent actions as `pick(x,y)` and `place(x,y)` with continuous `(x,y)` coordinates; this representation is integrated into our latent transition model through action attention and iterative refinement by a stack of residual graph neural networks. Our action attention module compares action information with the latent state of each object in order to predict the probability that an object is affected by the action. Each node in a GNN then receives a different action based on the predicted attention weights. Since each GNN in our transition model has access to the action, the responsibility for deciding which object is affected by the action and modeling its effect on said object can be distributed across the transition model.

We train and evaluate our factored world model in a simulated environment involving a UR5 robotic arm manipulating blocks of various shapes (Figure 1). We instantiate two environments: Cubes includes six cubes arranged into five different structures, which take up to ten actions to build, and Shapes includes eight blocks of four types (cube, brick, triangle, roof), which are arranged into sixteen different structures (Figure 3). To test zero-shot generalization — the main result of our paper — we partition the tasks into training and generalization sets. We ensure that the world model never sees the goal state of any generalization task during training.

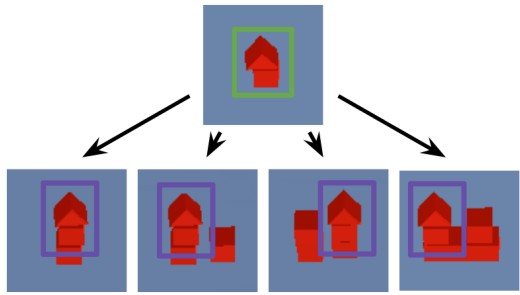

Figure 2: Example of generalization.

Nevertheless, our model transfers to the testing tasks in Cubes and Shapes with only a very minor drop in accuracy (Table 1). The mechanism behind successful zero-shot transfer is the ability of our model to understand the state of each object independently of others, and the ability to model sparse interactions. For example, if our model understands the interaction "stack a triangle on top of a cube" based on the first training task in Shapes (Figure 3, bottom row), it can use it when generalizing to four different held out tasks (Figure 2).

In summary, we contribute the following:

1. We develop a factored wold model for robotic pick-and-place tasks. This model does not require a known factored action-object association (as was used in related models), owing to its use of an action-attention network.

2. We demonstrate that the learned world model accurately predicts the outcomes of sequences of actions. Moreover, the model achieves zero-shot generalization to sequences of actions tasks and object configurations that were not seen during training, including sequences that are longer (up to 10 actions) than those in the training data (up to 10 actions). Finally, we demonstrate transfer to a real-world dataset.

## 2    FACTORED WORLD MODEL FOR ROBOTIC PICK-AND-PLACE

The modeling task is to predict the effect of a sequence of actions $a^1, a^2, ..., a^T$ in an environment that is initialized to a state $s^0$. We will assume a setting in which the state of the world is represented as an image, which has been pre-processed into a factorized state $s = \langle s_1, s_2, ..., s_K \rangle$ in which each $s_i$ is an image centered on the $i$th object. The number of objects $K$ can vary between episodes. This postulated factorization can be implemented using an object detection module that predicts a bounding box for each object and a tracking module that corresponds bounding boxes across time. Both problems are well-studied for natural images in computer vision (Zhao et al., 2019; Greff et al., 2020).

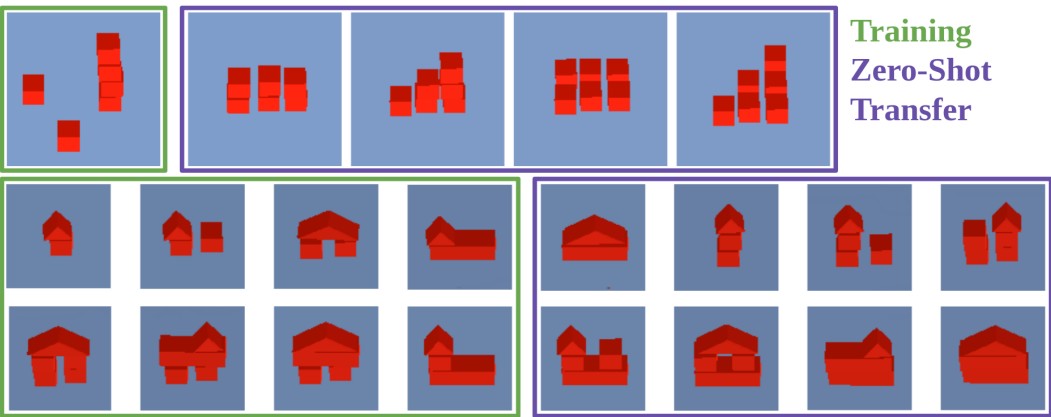

Figure 3: Breakdown of tasks in Cubes (top row) and Shapes (bottom row) environments. In Cubes, we train on the task of making a stack of four blocks and test on two wall stacking and two stair stacking tasks. In shapes, we instantiate 16 tasks with structures of height of three and a roof and top, and split them 50/50 training/testing. We provide expert demonstrations for training tasks and test zero-shot generalization in prediction on testing tasks.

Our goal is to learn a factorized latent representation $z = \langle z_1, z_2, ..., z_K \rangle$, along with a world model that predicts the state $z'$ that results from taking an action $a$ at state $z$. We will show that the learned latent code is low-dimensional; it represents object identity and position while abstracting over irrelevant features such as object color. Our world model is implemented using a graph neural network (Gori et al., 2005) that represents action-conditioned pair-wise interactions of latent factors. It is able to perform several rounds of message passing, each successively refining the prediction of the physics of picking and placing objects. We do not learn a mapping from $z$ to $s$ (i.e. the world model only predicts the future within its latent code); nevertheless, we show the latent transition model successfully solves prediction and action ranking tasks.

Both the state encoder and the latent transition model are equivariant to the permutation of factors. That is, we can provide objects in any order as long as it is consistent within an episode. Moreover, the model is agnostic to the number of objects present in a scene — no parameters in the model are dependent on the number of objects — although the weights of the latent transition model might overfit if the number does not vary during training. We describe our world model in detail in Section 2.1 and the training procedure in Section 2.2.

## 2.1 FACTORED VISUAL WORLD MODEL WITH ACTION ATTENTION

**Encoder.** Given a factored state $s = \langle s_1, s_2, ..., s_k \rangle$, the encoder $f_\phi$ processes each factor (image of an object) separately. That is, $z = \langle f_\phi(s_1), f_\phi(s_2), ..., f_\phi(s_K) \rangle$. The encoder uses three convolutional layers followed by average pooling and three fully-connected layers. In principle, the use of average pooling enables the encoder to process images of any size, although we use a fixed image size. As described in Section 3, the per-object image $s_i$ includes both an RGB component and a coordinate grid component, which marks the location where the crop was taken (Figure 4).

**Transition Model.** The transition model $g_\theta$ accepts a factored latent state $z = \langle z_1, z_2, ..., z_K \rangle$, $z_i \in \mathbb{R}^{D_z}$ and an action $a \in \mathbb{R}^{D_a}$. It predicts the next latent state by outputting a residual:

$$\hat{z}^{t+1} = z^t + g_\theta(z^t, a^t). \tag{1}$$

We implement $g_\theta(z^t, a^t)$ as a stack of graph neural network layers with skip connections. Denoting the $i$th graph neural network layer as $\text{GNN}_i$ and the $i$th intermediate representation as $y_i$, the computation with $L$ layers is as follows:

$$y_1 = z + \text{GNN}_1(z, a)$$
$$y_2 = y_1 + \text{GNN}_2(y_1, a)$$
$$...$$
$$g_\theta(z, a) = y_{L-1} + \text{GNN}_L(y_{L-1}, a) \tag{2}$$

Our implementation of individual graph neural network layers follows Kipf et al. (2020): the network models pairwise interactions between latent state factors (corresponding to objects in the environment) using a fully-connected node network $h_n$ and an edge network $h_e$. Both are implemented as MLPs with one hidden layer. The edge network outputs an embedding for each directed edge $e_{i,j} = h_e(z_i, z_j)$. The edge embeddings are then aggregated using the node network in order to update the state of each node. Finally, the graph neural network outputs a vector of updated factors $z' = \langle z'_1, z'_2, ..., z'_K \rangle$ with

$$z'_i = h_n \left( z_i, a, \sum_{j \neq i} e_{j,i} \right). \tag{3}$$

The summation over edges in the input of the node network assures permutation equivariance, an important property that aids generalization to novel combinations of factors. Given a permutation of factors $\pi$, we have

$$\text{GNN}(\pi(z), a) = \pi(\text{GNN}(z, a)). \tag{4}$$

The permutation equivariance property also holds for a stack of GNN layers as well as the state encoder and action attention.

**Action Attention.** When picking or placing objects in our experiments, the robotic arm and gripper will usually interact with only one or a small number of objects. The transition model needs to learn which latent factors change and which stay the same when an action is performed. Previous work simplified this problem by assuming that the action space is also factorized (Sanchez-Gonzalez et al., 2018; Veerapaneni et al., 2019; Kipf et al., 2020), which is to say that actions are performed relative to individual objects or parts. This assumption is not realistic in typical robotics applications, where actions correspond to motor primitives that cannot trivially be associated with effects on individual objects. We use `pick(x,y)` and `place(x,y)` actions with continuous `(x,y)` coordinates.

We overcome this limitation by introducing an action attention layer. Instead of the action being the same for each node in the graph neural network, we create $K$ transformations of the action that are attenuated based on how likely they are to interact with each latent factor. The mechanism follows single-head self-attention (Vaswani et al., 2017) with the following key $k$, query $q$ and value $v$:

$$\begin{aligned} k &= \langle j_k(z_1), j_k(z_2), ..., j_k(z_K) \rangle, \\ q &= j_q(a), \\ v &= j_v(a). \end{aligned} \tag{5}$$

Here, $j_k$, $j_q$ and $j_v$ are Multi-Layer Perceptrons with one hidden layer. The factored action space $a' = \langle a'_1, a'_2, ..., a'_K \rangle$ is composed of $v$ weighted by attention weights $\alpha_i$ with $i \in \{1, ..., K\}$:

$$\begin{aligned} \alpha_i &= \text{softmax}(k_1^T q, k_2^T q, ..., k_K^T q)_i, \\ a'_i &= \alpha_i v. \end{aligned} \tag{6}$$

Note the new actions are all equal up to a multiplicative factor. Figure 7 visualizes $\alpha$.

## 2.2 Learning by Contrastive Loss

We train the encoder and latent transition model using a single-step contrastive loss. We use a contrastive loss with a single positive and negative example (Kipf et al., 2020). Given a transition $\langle s^t, a^t, s^{t+1} \rangle$, we encode the current and next state ($z^t = f_\phi(s^t)$, $z^{t+1} = f_\phi(s^{t+1})$, $\hat{z}^{t+1} = z^t + g_\theta(z^t, a^t)$) and make a prediction about the next latent state. The contrastive loss minimizes the distance between the real and predicted next latent state, while maximizing the encoding distance between the current state and a negative state ($\bar{z} = f_\phi(\bar{s})$) up to a margin $\gamma$:

$$L(z^t, z^{t+1}, \hat{z}^{t+1}, \bar{z}) = \frac{1}{2K\sigma^2} \sum_{i=1}^{K} \left\| z_i^{t+1} - \hat{z}_i^{t+1} \right\|_2^2 + \max\left\{ 0, \gamma - \frac{1}{2K\sigma^2} \sum_{i=1}^{K} \left\| z_i^t - \bar{z}_i \right\|_2^2 \right\}. \tag{7}$$

The negative state $\bar{s}$ is sampled by randomly permuting a training batch of current states. It is a proxy for sampling a random state from the entire dataset (without needing to increase the batch size). Intuitively, we want the encoder to capture the minimum information required to distinguish a randomly sampled pair of states while enabling the latent transition model to be accurate.

## 3 EXPERIMENTS

Our empirical evaluation focuses on pick-and-place robotic manipulation tasks performed in simulation as well as on a real-world robotic arm (Figure 1 and Figure 8). The environments involve a robotic arm manipulating objects of four shapes (cube, brick, triangle, roof) in a $30 \times 30$ cm workspace. The agent controlling the arm uses pre-defined `pick` and `place` motion primitives: it chooses a particular $(x, y)$ location (continuous action space) in the workspace in which the arm executes a top-down pick or place action depending on if it is holding an object.

The state of the environment is captured by two RGB cameras pointed at the workspace and a third RGB camera that captures the content of the robotic gripper. We assume access to a bounding box for each object. The factored state $s = \langle s_1, s_2, ..., s_K \rangle$ is created by cropping the contents of each bounding box with added padding and resizing the resulting image to an $18 \times 18$ square. We add bounding box coordinates into the cropped image in the form of four additional coordinate grids channels (Figure 4, Section A.1). Hence, each view of each object results in an $18 \times 18$ image with 3 RGB and 4 coordinate grid channels. Note that we only require object bounding boxes within the images taken by side-viewing cameras. We do not need the ground-truth $(x, y, z)$ object positions in order to generate the factored state. We concatenate the two views of each object channel-wise, creating a 14-channel image for each object. If an object is held by the robotic gripper, we replace the two views with two images of the gripper — one from the front and one from the side. These two images can be captured by a single camera by rotating the robotic arm in between images. The coordinate grids for hand images are set to zero.

The environment poses a difficult exploration problem: a sequence of random actions is unlikely to create a large structure without knocking it over. A model-free agent with a random exploration policy fails to learn to build structures involving more than two objects (Biza et al. (2021), Table 2, "DQN RS"). Therefore, we collect a dataset of demonstrations from an experiment with added randomness. We evaluate both the ability of our model to fit tasks with expert trajectories as well as the ability to generalize to a set of held-out tasks *without fine-tuning* (zero-shot transfer, Figure 3).

We aim to answer the following questions:

- Can the Factored World Model fit physics of robotic manipulation? (Section 3.1)
- Does permutation equivariance in the Factored World Model facilitate generalization to unseen tasks? (Section 3.2)
- What is the contribution of the individual components of our model to its performance? Which component of the factored state is the most important? (Section 3.3)
- Can we use a model trained in simulation to make prediction by trajectories generated by a physical robot? (Section 3.4).

### 3.1 LEARNING ROBOTIC MANIPULATION DYNAMICS

**Setup.** To train the Factored World Model, we collect datasets from the training tasks in the Cubes and Shapes environments (Figure 3). Each dataset consists of 200k transitions collect by an expert with added randomness, see Appendix A.2 for details. The Factored World Model learns for 200 epochs using contrastive learning, and is subsequently evaluated both on the training and testing tasks. We use a block position prediction metric to evaluate the quality of the learned representation as well as the latent transition model. We also evaluate the model in a setting where it predicts the outcomes of several action sequences. The action sequences are similar, but only one of them reaches the goal state of a given task. By predicting the outcome of each sequence, the model guesses which one it is. We use this setting as a proxy for planning.

Block position prediction (RMSE): After training, we freeze the model and train an additional decoder to predicts the $(x, y, z)$ position of the center of each object from its latent representation. The decoder, a Multi-Layer Perceptron with two hidden layers, is trained for five epochs. It is trained using additional supervision (pairs of states and position labels) not available during training of the Factored World Model. We evaluate both the model's ability to represent block positions in the current state ($t = 0$) and its ability to make predictions for trajectories ($t > 0$). We report the root mean squared error in centimeters. The quantity predicted in this setting is different than the bounding box coordinates provided to the model as input. The bounding box coordinates index into a flat 2D images; the model needs to account for perspective to predict the `(x,y,z)` positions of objects.

| Method | Cubes–Test Set | | Shapes–Test Set | |
|---|---|---|---|---|
| | RMSE (cm) | Hits@1 (%) | RMSE (cm) | Hits@1 (%) |
| RPIN (Qi et al. (2021)) | 20.0 | 0 | - | - |
| C-SWM (Kipf et al. (2020)) | $11.39_{\pm 0.02}$ | $0_{\pm 0.0}$ | - | - |
| FWM-AE | $2.01_{\pm 0.46}$ | $60.7_{\pm 15.6}$ | $\mathbf{0.67}_{\pm 0.32}$ | $65.1_{\pm 20.2}$ |
| **FWM (our)** | $\mathbf{1.46}_{\pm 0.23}$ | $\mathbf{98.3}_{\pm 0.6}$ | $0.82_{\pm 0.14}$ | $\mathbf{78.3}_{\pm 6.3}$ |
| FWM - 1 GNN Layer | $1.78_{\pm 0.09}$ | $83.5_{\pm 2.1}$ | $0.81_{\pm 0.08}$ | $57.4_{\pm 4.1}$ |
| FWM - 1 GNN, No Att. | $3.60_{\pm 0.56}$ | $43.1_{\pm 14.3}$ | $1.34_{\pm 0.05}$ | $46.2_{\pm 5.3}$ |
| FWM - No Attention | $1.63_{\pm 0.11}$ | $97.7_{\pm 0.5}$ | $0.89_{\pm 0.08}$ | $76.1_{\pm 2.4}$ |
| FWM - No RGB | $1.11_{\pm 0.07}$ | $97.3_{\pm 0.5}$ | $0.61_{\pm 0.06}$ | $78.0_{\pm 4.5}$ |
| FWM - No Coordinates | $11.64_{\pm 1.08}$ | $0_{\pm 0.0}$ | $9.56_{\pm 0.97}$ | $0_{\pm 0.0}$ |
| FWM - No Factorization | $10.19_{\pm 1.14}$ | $0_{\pm 0.0}$ | $6.19_{\pm 0.81}$ | $0_{\pm 0.0}$ |
| Method | Cubes–Zero-Shot | | Shapes–Zero-Shot | |
| | RMSE (cm) | Hits@1 (%) | RMSE (cm) | Hits@1 (%) |
| RPIN (Qi et al. (2021)) | 21.7 | 0 | - | - |
| C-SWM (Kipf et al. (2020)) | $9.15_{\pm 0.04}$ | $0_{\pm 0.0}$ | - | - |
| FWM-AE | $2.77_{\pm 1.82}$ | $37.9_{\pm 30.7}$ | $\mathbf{0.80}_{\pm 0.33}$ | $54.0_{\pm 26.5}$ |
| **FWM (our)** | $\mathbf{1.88}_{\pm 0.20}$ | $\mathbf{98.4}_{\pm 0.5}$ | $1.02_{\pm 0.16}$ | $\mathbf{77.4}_{\pm 6.1}$ |
| FWM - 1 GNN Layer | $2.05_{\pm 0.17}$ | $95.0_{\pm 0.8}$ | $1.02_{\pm 0.08}$ | $56.5_{\pm 2.9}$ |
| FWM - 1 GNN, No Att. | $5.89_{\pm 0.71}$ | $33.2_{\pm 18.5}$ | $1.60_{\pm 0.05}$ | $51.0_{\pm 5.2}$ |
| FWM - No Attention | $2.22_{\pm 0.24}$ | $97.7_{\pm 0.4}$ | $1.09_{\pm 0.08}$ | $75.7_{\pm 3.0}$ |
| FWM - No RGB | $1.39_{\pm 0.38}$ | $98.2_{\pm 0.4}$ | $0.76_{\pm 0.07}$ | $79.8_{\pm 4.5}$ |
| FWM - No Coordinates | $9.82_{\pm 4.11}$ | $0_{\pm 0.0}$ | $9.81_{\pm 0.95}$ | $0_{\pm 0.0}$ |
| FWM - No Factorization | $8.39_{\pm 1.10}$ | $0_{\pm 0.0}$ | $6.79_{\pm 0.82}$ | $0_{\pm 0.0}$ |

Table 1: Comparison between Factored World Models, baselines and ablations in Cubes and Shapes environments. We evaluate the models both on the tasks they were trained on (top section) and on unseen tasks without additional fine-tuning (bottom section). While the bottom section reports results for generalization to unseen tasks, reaching high scores in the top section requires in-distribution generalization to known structures built in new positions in the workspace. Hence the terms "Test Set" and "Zero-Shot". We report block position error (RMSE, the lower the better, unbounded) and action sequence ranking score (Hits@1, the higher the better, bounded 0 - 100) (Section 3.1). Each model was run with 8 random seeds and we report means and 95% confidence intervals.

Action sequence ranking (Hits@1, MRR): We start with an optimal action sequence that achieves a goal state of a particular task. Then, we generate ten other action sequences where each action is perturbed by noise with magnitude $\epsilon$. The noise is added to each action in the sequence by drawing $\epsilon \sim \text{Unif}[0, \epsilon]$ and $\theta \sim \text{Unif}[0, 2\pi]$ that are then combined to create a 2D vector in the direction of $\theta$ with a length of $\epsilon$ that is added to the (x, y) coordinate of each action. Each perturbed action sequence must not achieve the given task; otherwise, we re-sample noise. The model observes the starting state and predicts the final state of each action sequence. We also encode the final state of the correct action sequence and compare its distance to the *predicted* final states of all action sequences. Hits@1 report the fraction of times the model's prediction of the final state of the correct action sequences was closer to the encoded final state than the predictions for all of the incorrect action sequences. Since the incorrect action sequences are chosen to be a small distance from the correct one, a model making random predictions is not expected to reach Hits@1 of 1/11. In fact, the model needs to be fairly competent to get a non-zero score.

**Results.** We report results for training tasks in Table 1 top section, Figure 5 left and Figure 6 left. The Factored World Model reaches low block prediction error (1.5 cm for Cubes and 0.8 cm for Shapes; the size of a cube is 3 cm for comparison) and high action ranking score (98% for Cubes and 78% for Shapes). We compare to an autoencoder baseline (FWM-AE) and the Region Proposal Interaction Network (RPIN, Qi et al. (2021)) with minor changes. For FWM-AE, we add a decoder to our model and train with an autoencoding loss (specifically the Embed to Control loss Watter et al. (2015)) instead of contrastive learning. To adapt RPIN, a factored video prediction method, we append an (x, y, pick/place) action to the input of the Prediction module in their Convolutional

Interaction Network. We recreate our dataset in the same format as their Shape Stacking experiment ($224\times224$ images with a bounding box for each object) and we use the same hyper-parameter.

Compared to an autoencoding loss (FWM-AE), contrastive learning (FWM) succeeds both in Cubes and Shapes, whereas the autoencoding loss fails to fit Cubes. In Shapes, autoencoding outperforms contrastive loss in terms of block position prediction, but underperforms in action sequence ranking. We believe the difference can be explained by the focus of the two different losses: autoencoding loss directly incentivizes the model to reconstruct the state of the environment (which include bounding box coordinates in the form of a coordinate grid), whereas contrastive loss focuses on learning compact state representations predictive of the future.

Our adaptation of RPIN to action-conditioned sequence prediction fails to learn a dynamics model. The encoder of RPIN can capture some position information: it reaches around 6 cm RMSE for the task of predicting ground-truth object coordinates without forward modeling. However, the forward model fails, reaching 20 cm RMSE.

We further study the block prediction error in Figure 6. Figure 6 reports block prediction error for noisy trajectories generated by an expert policy with added randomness, whereas Table 1 reports errors for optimal trajectories that reach the goal state of each of the training (and generalization) tasks. In Shapes (Figure 6 right), we see a reversal compared to Table 1, as contrastive loss outperforms autoencoding by a small margin. This result suggests that the transition model learned by autoencoding is not as accurate for blocks falling and colliding with each other.

Finally, we plot action ranking Hits@1 as a function of noise level $\epsilon$ for Shapes in Figure 5 left. Contrastive loss outperforms autoencoding for all noise levels with the area under the curve being 84% for FWM and 64% for FWM-AE.

### 3.2 Zero-Shot Generalizing to Novel Tasks

**Setup.** We further evaluate models trained in Section 3.2 on tasks unseen during training (Figure 3). We specifically make sure that the training datasets do not contain a single example of a goal state of the novel tasks. In Cubes, the training task is building a 4-stack and the novel tasks are building a wall, stairs, a wall on ground and stairs on ground. We found that the training data collection policy never solves the novel tasks (we use a separate policy to collect evaluation trajectories for testing tasks). In Shapes, we train on half of the possible structures with a height of three and roof on top, and test generalization on the other half. In this case, the data collection policy for the eight training tasks solves one of the testing tasks around once every 50 episodes (due to added randomness) and we delete these episodes. We report block position error and action ranking Hits@1 described in Section 3.1. We do not perform any fine-tuning on the novel tasks in this experiment.

**Result.** In both Cubes and Shapes, FWM transfers to the novel tasks with only a minor decrease in performance (block position error increases by 0.4 cm in Cubes and 0.2 cm in Shapes and action sequence ranking Hits @1 remain almost unchanged). We consider this to be an important demonstration of the generalization properties of permutation equivariant models. In contrast, only limited generalization and transfer properties have been shown in prior work (Li et al., 2020; Biza et al., 2021). FWM-AE suffers a large decrease in action sequence ranking while still outperforming FWM on block position prediction in Shapes. We plot action ranking error as a function of $\epsilon$ for Shapes in Figure 5 right; we see a pattern analogous to results for training tasks (Section 3.1). The areas under curve are 81% for FWM and 48% for FWM-AE.

### 3.3 FWM Ablations and Attention Visualization

We report the results of the following ablations in Table 1: removing action attention (No Attention and 1 GNN, No Att.), using only one graph neural network layer (1 GNN Layer), only showing bounding box coordinates as inputs (No RGB), only showing RGB images as inputs (No Coordinates) and using a monolithic latent transition model (No Factorization).

We did not find a significant difference when using FWM with and without action attention. As shown in Figure 7 first row and third row, FWM learns interpretable attention weights. During a pick action, the attention weights reflect the object being picked up as well as any object that is nearby. The attention module has an interesting behavior during place actions: it gives a weight of

zero to the object being placed, but attends to all objects that are underneath or next to the place location. For example, in the top-right image in Figure 7, the green blocks is being placed on top of the red, blue and black blocks. The attention module only attends to these three blocks with the highest weight assigned to the red blocks, which is directly below the object being placed.

Why is there no quantitative increase in performance even though the attention module functions properly? We hypothesize that what the attention module does *explicitly* the four GNN layers in FWM can learn to do *implicitly* using the node and edge networks. To test this hypothesis, we train a model with a single GNN layer (commonly used in prior work) with and without attention. We assume that the reduced capacity of the model will require it to use the attention module. Here, action attention leads to large quantitative improvements. E.g. not using attention more than doubles block prediction error in Cubes. Furthermore, qualitative analysis shows that the attention module behaves differently with 1 GNN layer compared to 4 GNN layers. With 1 GNN layer, the module strictly attends to the object being manipulated (Figure 7, second and fourth row), during both `pick` and `place`, while ignoring objects that are nearby.

A second interesting result in our ablation studies is the comparison of FWM with and without RGB image. The No RGB baseline only has access to the coordinates of the bounding boxes for each object. We see no statistically significant different between these two settings. Since the RGB baseline has access to bounding box coordinates from two different cameras, it can use this information to triangulate where an object is located in the workspace. We believe more complex object interaction (e.g. packing a bin with household objects, building from LEGO) would put higher emphasis on images. We leave these experiments for future work.

No Factorization and No Coordinates ablations show that both of these components are vital to the Factored World Model. Without coordinate grids, the model perceives images of objects without any information about where they are located in the environment. Without object factorization, the monolithic latent transition model is unable to generalize beyond the specific arrangements of blocks seen in the training tasks. Here, we mean in-distribution generalization to building known structures in novel locations of the workspace. Naturally, the monolithic model also fails in zero-shot transfer. Finally, we find stacking of four residual GNNs leads to large improvements compared to a single GNN layer. In Shapes, the action ranking Hits@1 increase by 19% with four GNN layers.

### 3.4 TRANSFER TO A REAL-WORLD DATASET

**Setup.** We collect evaluation trajectories on a physical robot (Figure 8) for the five cube stacking tasks shown in Figure 3, top row. The real-world dataset includes bounding boxes (generated by segmenting cubes by color) and ground-truth block positions annotated by hand with the help of a depth camera. We do not train our model on the real-world dataset, since we only collect 20 episodes for each task. Instead, we change the background and object colors in simulation to match the real-world images. In all cases, we train on the task of stacking four blocks on top of each other and transfer to other four tasks in the Cubes environment. That is, we test the ability of our model to transfer from building a tower of four blocks in simulation to building stairs, walls, etc. in the real world without any additional fine-tuning.

**Result.** We compare results from simulation and the real-world in Table 2. On average, the block position prediction error increases by 0.4 cm as we transfer from simulation to the real-world dataset. Our model appears to struggle with tasks that involve stacking blocks on top of each other in the real world compared to simulation. Conversely, there no decrease in performance for Wall Ground (WG) and Stairs Ground (WG), where all blocks are placed on the ground.

### 4 RELATED WORK

**World Models with Object Factorization.** Action-conditioned object-factorized world models have been explored in toy object environments and Atari games (Kipf et al., 2020; Huang et al., 2020) as well as in robotic manipulation (Veerapaneni et al., 2019) and controlling billiard balls (Kossen et al., 2020). We build upon C-SWM (Kipf et al., 2020), which has been further extended by (Huang et al., 2020) to handle identically looking objects (one of the assumptions of C-SWM is that each object is distinct). Veerapaneni et al. (2019) used a factored transition model to learn and plan for stacking up to three cubes. Different from these works we consider a simplified setup wherein

| Training | Evaluation | RMSE (cm) | | | | | |
|---|---|---|---|---|---|---|---|
| | | 4-Stack | Wall | Stairs | WG | SG | Avg |
| Sim 4-Stack | Sim All | $1.46_{\pm 0.23}$ | $1.76_{\pm 0.18}$ | $2.14_{\pm 0.25}$ | $2.14_{\pm 0.25}$ | $1.90_{\pm 0.21}$ | $1.88_{\pm 0.20}$ |
| Sim2Real 4-Stack | Real All | $2.33_{\pm 0.4}$ | $2.97_{\pm 1.6}$ | $2.40_{\pm 1.5}$ | $1.72_{\pm 0.6}$ | $1.81_{\pm 1.5}$ | $2.25_{\pm 0.8}$ |

Table 2: Block position predictions results for a real-world evaluation dataset. The first row is a breakdown of results previously reported in Table 1 (FWM). The second row reports results for a model trained in simulation and evaluated on trajectories captured on a physical robot. 4-Stack, Wall, Stairs, Wall Ground (WG) and Stairs Ground (SG) correspond to the tasks shown in Figure 3.

our model does not perform symmetry breaking or object discovery from raw visual features, since we assume our environment to be already factored.

Object-factored world models are also commonly used in the context of physics prediction in videos. Janner et al. (2019); Veerapaneni et al. (2019) demonstrated generalization in a Tetris-like environment, where a world model predicts the outcome of dropping blocks from a height. Ye et al. (2019) studied learning dynamics of pushing objects with a robotic arm in the real world. RPIN (Qi et al., 2021) demonstrated state-of-the-art results in the tasks of 2D physics modeling (PHYRE benchmark, Bakhtin et al. (2019)), modeling billiard balls and modeling videos of falling stacks of blocks. As we showed in our comparison, models like RPIN cannot be naively applied to our pick-and-place tasks, both due to the problem of including action information at the right point in a transition model and due to the large changes between subsequent states in our environment (object disappearing from the environment vs. a video of an object being slowly picked up by a robotic hand).

Object factorization is an instance of a general factored Markov Decision Process, which has been studied in the context of policy search (Guestrin et al., 2003), factor discovery (Jonsson & Barto, 2005) and MDP abstraction (Ravindran, 2004; Wolfe & Barto, 2006). In fact, factored MDP works often include examples where individual factors are objects (e.g. Tower of Hanoi in Ravindran (2004)). Early works specifically on object factorization include Wolfe (2006); Diuk et al. (2008).

**Model-based Learning for Robotics.** One prominent line of work focuses on optimal control or policy search in low-dimensional state space. If such space is directly available, PILCO fits a Gaussian Process to real-world robot dynamics and performs policy search through gradient descent in the model (Deisenroth & Rasmussen, 2011). Successful cart-pole swing and robotic unicycle balance is achieved within a few seconds of online learning. In the case of image states, a low-dimensional latent space can be learned by variational autoencoders (see Lesort et al. (2018) for a survey). Watter et al. (2015) learn a latent state space together with a locally linear transition model, enabling efficient inference in optimal control methods. Watter et al. (2015) demonstrate successful control of pendulum, cart-pole and a simplified robotic arm operating in 2D. Follow-up works add additional constraints to the framework in order to improve dynamics model accuracy and robustness (Karl et al., 2017; Banijamali et al., 2018; Levine et al., 2020).

Other works have used models of the world to accelerate learning for robotic manipulation tasks. In Nair et al. (2018), a VAE, trained on image observations, is used for sampling goals during training and providing a reward signal via distance in the latent space. This method achieved similar sample complexity to a state-based method on a robotic pick-and-place task. In Nasiriany et al. (2019), a challenging robotic pushing task was performed by searching for sequences of subgoals in the latent space of a VAE that could be followed by a learned policy.

## 5 CONCLUSION

The results in this paper demonstrate that factorized world models can successfully be adapted to manipulation tasks with continuous state and action spaces by introducing an action-attention model to isolate the effects of actions on individual objects. The resulting models are able to learn transition dynamics that generalize to previously unseen sequences of actions and configurations of objects at test time, and are able to predict outcomes at moderate time horizons of up to 10 actions. This represents a significant step in applying these models in practical robotics tasks. An immediate future line of work is to integrate these world models with planning algorithms, and evaluate the ability of the resulting planner to generalize to tasks that were unseen during training.

REPRODUCIBILITY STATEMENT

We will open-source our code with the camera-ready version of our paper. We will include our simulated PyBullet environments, dataset generation scripts, Pytorch model definitions and training scripts.

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

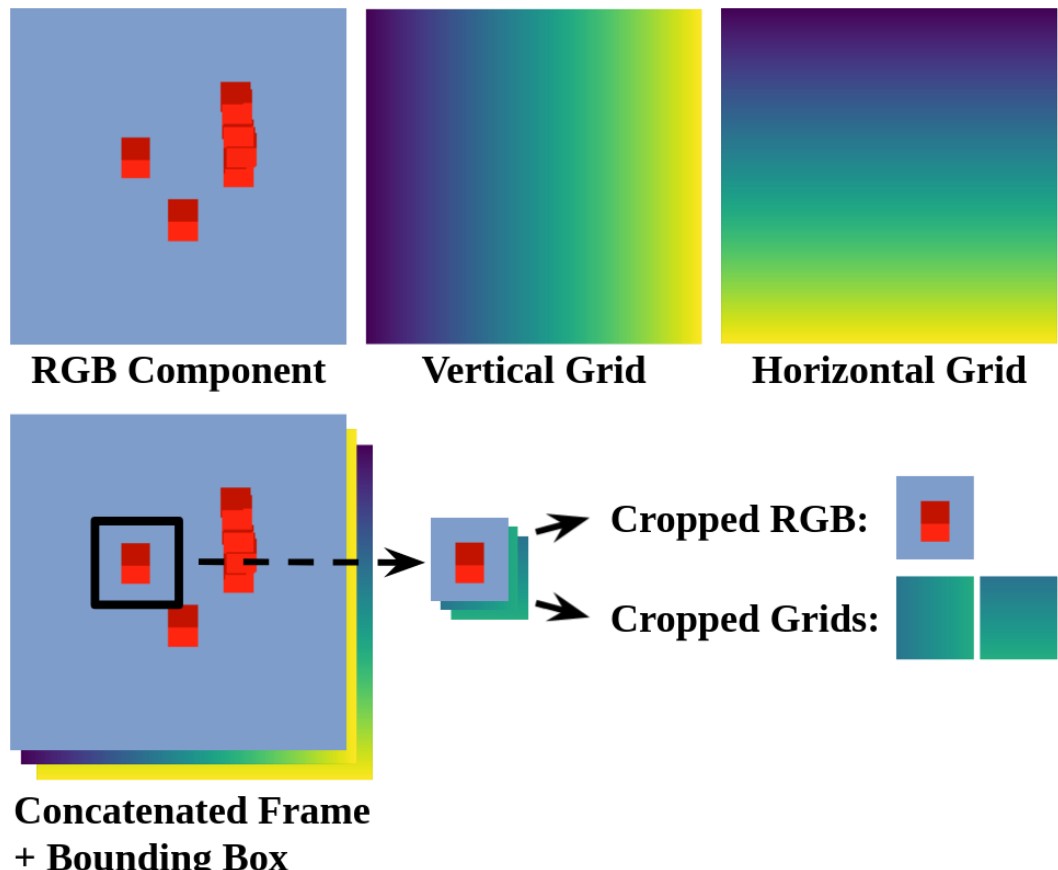

Figure 4: Schema of our pipeline for creating cropped object images. We concatenate an RGB image with horizontal and vertical coordinate grids. Using a bounding box with added padding, we crop an 18×18 image from both the RGB component and the coordinate grid components. By observing the coordinate grids, our agent known where *in the image* the object was cropped. Note that the coordinate grids are derived from object bounding boxes, not the actual (x,y,z) object positions in the environment. Hence, we do not need to know the ground-truth object positions in order to generate our factored states. We add two additional coordinate grids by mirroring the vertical and horizontal grids (similar to positional encodings in Locatello et al. (2020)).

# A   EXPERIMENT DETAILS

## A.1   ENVIRONMENT DETAILS

Our PyBullet simulation consists of a UR5 robotic arm with a Robotiq gripper operating over a 30×30 cm workspace. In comparison, the size of a cube is 3×3 cm. The environment is captured from two sides by two cameras. Each camera produces a 90×90 image. PyBullet also provides a segmentation mask for each object, which we use to create bounding boxes. We first draw the smallest rectangular box that captures the whole segmentation mask and add 4 px symmetric padding to it. If the bounding box is smaller than 18×18 px, then they are padded up to the minimum size. We crop the RGB image and four coordinate grids inside of the bounding box and resize them to an 18×18 image. The aspect ration of the RGB image gets corrupted in this step, but the model can still capture the size of each object based on the coordinate grids. We have a vertical and a horizontal coordinate grid that traverses the interval of [-1, 1] either left-to-right or up-to-down. We also create two additional coordinate grids that traverse the interval right-to-left and down-to-up.

**Block colors.** We use a single block color (red) for all experiments in simulation. In Figure 7, blocks are colored only for the purpose of matching blocks to attention weights. All blocks are red in the

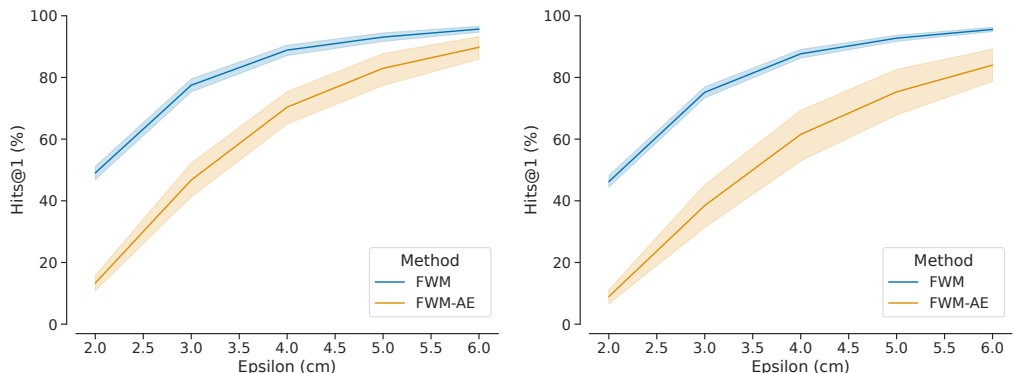

Figure 5: Action sequence ranking Hits@1 conditioned on action noise $\epsilon$. The noise level controls the degree to which negative action sequences differ from the position one (Section 3.1). Left: results for training tasks; right: results for zero-shot generalization. We compare Factored World Model against baselines from Table 1. Means and 95% confidence intervals over 5 random seeds are reported.

actual states passed into the model. In the real-world experiments, we assign each block a different color so that we can get bounding boxes using color segmentation (Table 2). We use different colors both in the Sim2Real dataset (generated in simulation) and the Real dataset (generated on a physical robot),

## A.2 EXPERT POLICIES

**Experts for training datasets.** For Cubes, we have the following policy: with 70% probability either pick a random cube that is not covered by other objects or place a cube on top of any other cube. Otherwise, execute a pick or place action in a random coordinate. Additionally, 1 cm random noise is added to the (x,y) position of each action. The agent does not have to decide between pick and place actions: hand empty → pick, hand full → place.

For Shapes, we use trained SDQfD agents provided by Biza et al. (2021). With 80% probability we execute the actions predicted by the expert. Otherwise, during pick, we pick a random object with 50% and we execute a pick at a random location otherwise; during place, we place in a random position.

We collect 200k training transitions for Cubes and Shapes.

**Experts for generalization datasets.** We simply collect optimal trajectories for evaluation and filter out any trajectory that does not reach the goal of each task.

## A.3 MODEL DETAILS

**Per-object Encoder CNN.** Conv2D($5\times5$ kernel size, 32 kernels, stride 2, padding 1) → BatchNorm (Ioffe & Szegedy, 2015) → ReLU → Conv2D($5\times5$ kernel size, 64 kernels, stride 2, padding 1) → BatchNorm → ReLU → Conv2D($5\times5$ kernel size, 64 kernels, stride 2, padding 1).

**GNN.** Both the node and the edge networks are MLPs with one hidden layer of size 512. Each layer, except the output layer, is followed by LayerNorm (Ba et al., 2016) and ReLU activation.

**Action Attention.** The key, value and query MLPs have a single hidden layer of size 512, the output size is also 512. LayerNorm and ReLU is used in the same way as in the GNN.

**Decoder for FWM-AE Baseline.** ConvTranspose2D($5\times5$ kernel size, 64 kernels, stride 2) → BatchNorm → ReLU → Conv2D($3\times3$ kernel size, 32 kernels, stride 2) → BatchNorm → ReLU → Conv2D($3\times3$ kernel size, 14 kernels, stride 1).

**Training.** We use the Adam (Kingma & Ba, 2015) optimizer with default parameters and a learning rate of $5e-5$. We train for 200 epochs (dataset size is 200k transitions) with a batch size of 256.

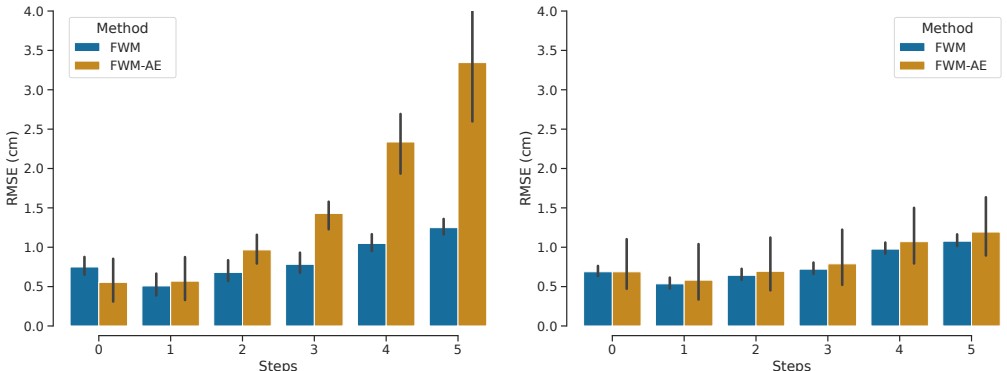

Figure 6: Block position prediction error as a function of the number of predicted time steps (Left: Cubes, right: Shapes). The validation set used in this figure contains noisy trajectories for training tasks, whereas the errors reported in Table 1 are for optimal trajectories that reach the goal of each task. We compare Factored World Model against baselines from Table 1. Means and 95% confidence intervals over 5 random seeds are reported.

# B  ADDITIONAL EXPERIMENTS

## B.1  C-SWM WITH HEURISTIC ACTION FACTORIZATION

We train the C-SWM model (Kipf et al., 2020) in the Cubes environment. Unlike FWM, C-SWM does not receive a factored state space; instead, it can choose what information is captured in each object slot. We make two changes to our environment to help C-SWM factor it: (1) we give each object a distinct color (Figure 9) so that the model can potentially learn color-specific filters and (2) we create a heuristically factored action space. The factored action space only provides action $a_i^t$ to the $i$th node in the C-SWM graph neural network if the $i$th object changed between state $s^t$ and $s^{t+1}$. Otherwise, the $i$th node receives a null action.

The model receives four images concatenated channel-wise: two images of the workspace and two images of the robot hand (which indicate if the robot is holding an object). We use a custom encoder with the following architecture: Conv2D($5\times5$ kernel size, 64 kernels, stride 2) $\rightarrow$ BatchNorm $\rightarrow$ LeakyReLU $\rightarrow$ Conv2D($5\times5$ kernel size, 64 kernels, stride 2) $\rightarrow$ BatchNorm $\rightarrow$ LeakyReLU $\rightarrow$ Conv2D($5\times5$ kernel size, 6 kernels, stride 1) $\rightarrow$ BatchNorm $\rightarrow$ ReLU. The output of the encoder is a $16\times16$ feature map for each object.

Across a range of learning rate, C-SWM does not learn to factor the state space of Cube stacking. By our metrics, C-SWM is on-par with an unfactored model (Table 1). Figure 9 visualizes the learned feature maps for each object slot: the maps follow an ABAB pattern, where the model appears to only distinguish between the robot holding or not holding an object. This pattern holds across episodes.

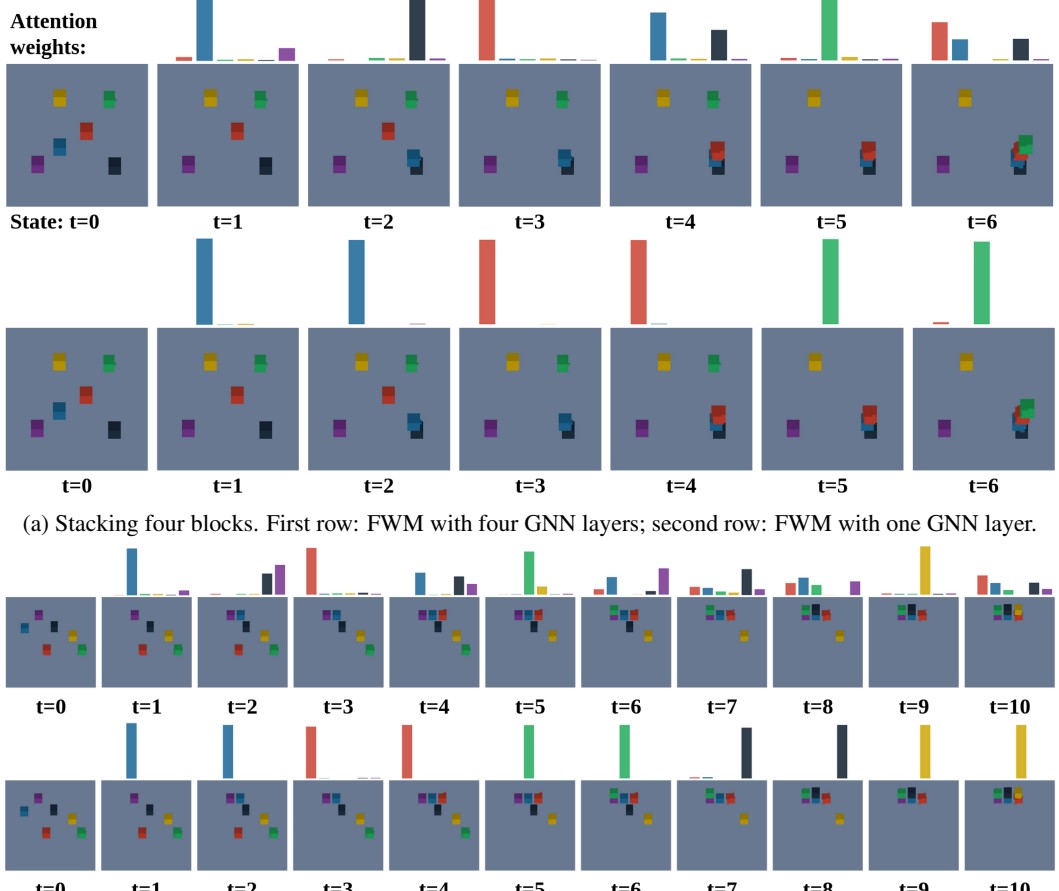

(a) Stacking four blocks. First row: FWM with four GNN layers; second row: FWM with one GNN layer.

(b) Building a wall. First row: FWM with four GNN layers; second row: FWM with one GNN layer.

Figure 7: Visualizing action attention weights for a sequence of building a stack of four blocks and a wall. Each bar in the histogram is associated with a particular object by color. (a) the agent executes a `pick` action at $t = 1, t = 3, t = 5$ and a `place` action at $t = 2, t = 4, t = 6$. (b) `pick` action at $t = 1, t = 3, t = 5, t = 7, t = 9$ and `place` action at $t = 2, t = 4, t = 6, t = 8, t = 10$.

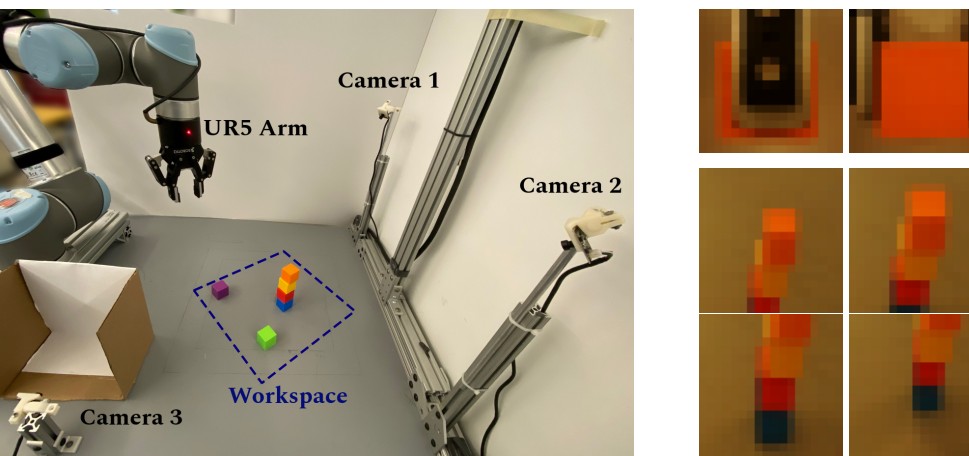

Figure 8: Left: our real-world experimental setup includes a UR5 arm with a Robotiq gripper, two RGB cameras facing the workspace (Camera 1 and 2) and one RGB camera to take an image of an object the robot is holding (Camera 3). The gripper moves inside of the box in the bottom-left corner after every successful pick action. Top-right: an in-hand image taken by Camera 3. Bottom-right: the four images represent a single factored state of the environment with a tower of four blocks. Each image is centered on one block starting from the top block going to the bottom. State factorization is explained in Figure 4.

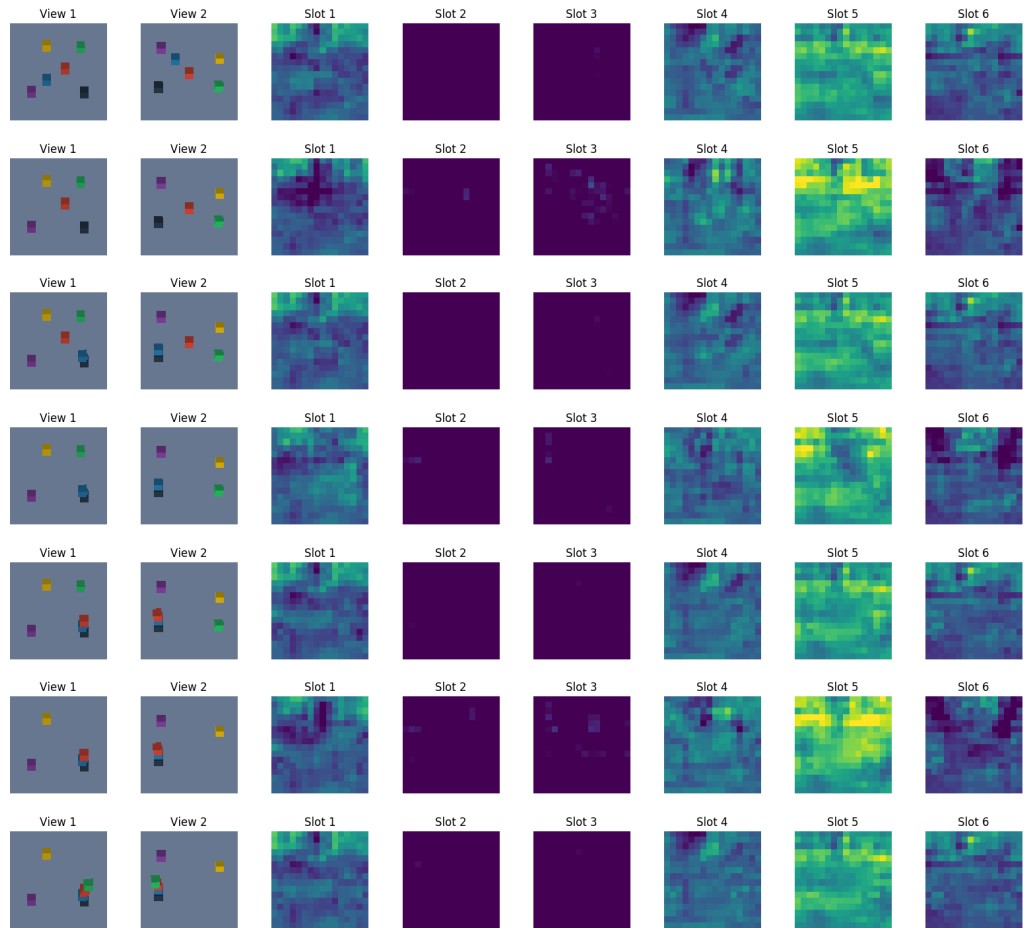

Figure 9: Visualization of feature maps learned by C-SWM with heuristic action factorization. The first two columns show the two views of the environment provided to the model. The next six columns show the $18\times18$ feature maps for each object slot given by the C-SWM object extractor. The colormap is scaled between 0 and 0.4.

