# OpenReview forum: "Factored World Models for Zero-Shot Generalization in Robotic Manipulation"
_ICLR.cc/2022/Conference — ICLR 2022 Submitted_

### Official Review · Reviewer_iwuh · 2021-11-01

**Correctness:** 1
**Technical Novelty And Significance:** 3
**Empirical Novelty And Significance:** Not applicable
**Recommendation:** 5
**Confidence:** 4

**Main Review:**

Strengths:
1) The paper is well written and is easy to understand in a go.
2) The action-attention module seems to be a novel idea especially if I think about a multi-agent scenario where it'd be useful to have such a module to direct the policy (or an accurate world model learning) and be sample-efficient (as opposed to just concatenating the action's latent to every object's latent state).

3) I appreciate the author(s) adding Figure 5 as it provides a  good visualization of the action-attention module.


Weaknesses:


4) Action Attention Module: Although I like the idea of having such a module, I am not convinced by the experiments performed to evaluate this. The task of pick and place is simple enough to implicitly figure out which object does an action correspond to -- this can be seen from Table 1, FWM and FWM - No attention. There is barely any improvement in the Hits@1 (and even the block's position if you consider the std. deviation along with the mean). This implies that not having action attention suffices for the network to figure out which object is the action intended on.

This is not to say that the action attention is useless. It is to say that for this specific task of pick and place, it is not very useful.

5) Claim about combinatorial generalization: The combinatorial generalization is due to the use of GNNs which is a core module of C-SWM [1]. Hence I do not agree with the claim that the zero-shot generalization shown in the paper is a contribution by FWM rather the author(s) just extends the claim made by C-SWM by showcasing it in a different environment (the robotic manipulation environment).

6) Zero-Shot generalization: Based on my points (5) and (6), it is important to make the distinction that the action attention does not contribute to the generalization (at least from the results shown on pick and place task in Table 1 on both Cubes and Shapes environment), rather it is the GNN that contributes to the zero-shot generalization. These two are often worded one after the other which makes the reader think that it is the action-attention that is major contributing factor to the generalization. Kindly make this clear in the text.

7) "No RGB" Experiment: There is no discussion about this experiment but in the environments considered by the author(s) all the objects are in red and hence RGB doesn't really play a role in the task. This is also shown clearly in Table 1, where the performance is almost similar (a slight increment/decrease on the RMSE/Hits@1) to FWM.

8) Real-world experiments? - The first line of the Experiments section says "Our empirical evaluation focuses on pick-and-place robotic manipulation tasks performed in simulation as well as on a real-world robotic arm", but I do not see any real-world robotic arm experiments. Could the author(s) clarify this?

9) Why only Robotic manipulation? - The action-attention module and (other components from C-SWM) are very generic, so why did the author(s) restrict themselves to only robotic manipulation environments? What about Atari games or other object-based RL environments? It would be interesting to see if action-attention plays an important role there (especially in getting to know the agent in Atari game a.k.a the most controllable agent).

10) In Fig 5, are the objects colored only for visualization purposes or is this the RGB input?

[Minor thing which I have not considered for rating the paper]

11) There are some places where the citations have not been mentioned (rather the literal word "citations" appear twice in the text). For example: pg 2(last line) and pg 3 (first paragraph, 5th line). This is being disregarded by me for evaluating the paper since it seems to be a genuine mistake caused by overlooking. I'd request to add the citations in the future version of the manuscript.

---
References

[1] Contrastive Learning of Structured World Models, Thomas Kipf et. al, ICLR 2020

**Summary Of The Paper:**

The paper proposes Object-Factored world models, a world model learned using contrastive loss, that tends to generalize over the number of objects in the scene. It also alleviates the assumption that previous world models hold about the object-action association and the author(s) propose an "action-attention module" that estimates the probability of an action affecting a particular object (using self-attention). Their experiments mainly focus on robotic manipulation tasks and they show generalization on unseen-test tasks with little drop in performance as compared to the training set.


**Summary Of The Review:**

Overall I feel that the paper lacks a concrete quantitative evaluation of the action attention module (which is the core contribution of the paper). The experiments don't prove that this module is helpful for robotic manipulation tasks (though it can be useful for some other tasks which the author(s) have to investigate further). Hence, I would like to vote for a weak rejection of the paper at this stage. I will be making a final decision after engaging with the author(s) post rebuttal.

---

> ### Author Response · Authors · 2021-11-20
> **Author Response**
>
> > Action Attention Module: Although I like the idea of having such a module, I am not convinced by the experiments performed to evaluate this. The task of pick and place is simple enough to implicitly figure out which object does an action correspond to -- this can be seen from Table 1, FWM and FWM - No attention. There is barely any improvement in the Hits@1 (and even the block's position if you consider the std. deviation along with the mean). This implies that not having action attention suffices for the network to figure out which object is the action intended on.
>
> This point was also raised by Reviewer WeZs and Reviewer 4oxK. We added more experiments and analysis of attention in Section 3.3 and Figure 7. To summarize, we see a large benefit to using action attention when we train FWM with one GNN layer. We hypothesize that our full model, which uses four GNN layers, does not need action attention, since it has an abundance of representation capacity. We further show that with both one and four GNN layers, action attention learns interpretable weights (please see Figure 7). These weights are useful for both model analysis and for downstream tasks.
>
> > Claim about combinatorial generalization: The combinatorial generalization is due to the use of GNNs which is a core module of C-SWM [1]. Hence I do not agree with the claim that the zero-shot generalization shown in the paper is a contribution by FWM rather the author(s) just extends the claim made by C-SWM by showcasing it in a different environment (the robotic manipulation environment).
>
> The main result of our paper is zero-shot generalization to data drawn from a different distribution (previously unseen block structures). This is different from the results presented in C-SWM, where the training and testing data are drawn from the same distribution. C-SWM does not have to generalize to new types of interactions (e.g. same sub-structures built at different heights).
>
> It was not clear to us a priori that graph neural networks would be able to perform the type of generalization shown in Figure 2 (our paper). Here, the model generalizes to the particular stack of two blocks being built at a different height and surrounded by different objects. The ability to ignore height or neighboring objects is not trivially given by permutation equivariance.
>
> > "No RGB" Experiment: There is no discussion about this experiment but in the environments considered by the author(s) all the objects are in red and hence RGB doesn't really play a role in the task. This is also shown clearly in Table 1, where the performance is almost similar (a slight increment/decrease on the RMSE/Hits@1) to FWM.
>
> We added a discussion of No RGB to Section 3.4.
>
> > Real-world experiments? - The first line of the Experiments section says "Our empirical evaluation focuses on pick-and-place robotic manipulation tasks performed in simulation as well as on a real-world robotic arm", but I do not see any real-world robotic arm experiments. Could the author(s) clarify this?
>
> We added real-world experiments to the revised version of our paper. Please see Section 3.4.
>
> > Why only Robotic manipulation? - The action-attention module and (other components from C-SWM) are very generic, so why did the author(s) restrict themselves to only robotic manipulation environments? What about Atari games or other object-based RL environments? It would be interesting to see if action-attention plays an important role there (especially in getting to know the agent in Atari game a.k.a the most controllable agent).
>
> We agree that our attention module could be used outside of robotic manipulation. We are currently looking at adding the attention module to C-SWM and testing it on datasets collected in Atari games.
>
> > In Fig 5, are the objects colored only for visualization purposes or is this the RGB input?
>
> Yes, the actual objects are all red. The object colors are added so that they can be matched with the bar charts. We added details about object colors at the end of Appendix A.1 in the revised paper.
>
> > There are some places where the citations have not been mentioned (rather the literal word "citations" appear twice in the text). For example: pg 2(last line) and pg 3 (first paragraph, 5th line). This is being disregarded by me for evaluating the paper since it seems to be a genuine mistake caused by overlooking. I'd request to add the citations in the future version of the manuscript.
>
> We have fixed this problem. Thank you for pointing it out.

---

### Official Review · Reviewer_4oxK · 2021-11-02

**Correctness:** 3
**Technical Novelty And Significance:** 2
**Empirical Novelty And Significance:** 2
**Recommendation:** 5
**Confidence:** 3

**Main Review:**

Strengths:
 - The authors demonstrate that factored world models can indeed propagate multi-object structured relation modeling in robotic manipulation scenarios while enabling zero-shotting in unseen object/action configurations.
 - The addition of the attention module as well as stacking multiple GNN layers evidently increases  (although marginally and not always) the performance of the model across both tasks and evaluation metrics
 - The proposed factorization of the state representation is relatively straightforward to implement in the context  of robotic manipulation applications, as actuators are most commonly trained in simulation environments that  come packed with object segmentation tools (e.g. segmentation masks in Pybullet)

Weaknesses:
 - Even though this work seems to be an extension of the factorized world model proposed by Kipf et al. (2020) there are no direct comparisons given with their model in the experiments section.
 - The authors claim they wish to explore the potential of factored world models in generalizing to unseen tasks, but in the concerned environments, new "tasks" simply correspond to novel sequences of picking and placing actions (i.e. novel configurations), something previous works that impose permutation equivariance already demonstrate that they can achieve. This work's novelty focuses on the application of an attention mechanism and stacking multiple GNN layers. Both additions provide minor performance boosts (and not robustly across all experiments, as seen in rows 3, 4, and 5 of the bottom part of Table 1).
 - A few grammatical mistakes and forgotten to actually cite works in the first two paragraphs of Section 2.
- Inconsistent presentation of figures and tables throughout the text.

**Summary Of The Paper:**

In this paper, the authors expand on previous works on contrastive learning with factored world models (specifically the C-SWM model by Kipf et al. - 2020) and apply their model for the task of predicting the dynamics of object manipulation with a continuous action space. Such previous works have shown that by leveraging residual graph neural networks (GNNs) to contextualize the learned latent states with possible actions, the model learns representations that are equivariant to permutation of objects in the scene, thus allowing scaling to a high number of objects regardless of the combinatorial explosion of the state space. Unlike previous works, where new objects are discovered by the model from raw pixels, the authors here use the simulation environment to segment objects appearing in the scene and factor them as part of the state representation. Another difference is that instead of factorizing the action space, they use two action primitives for picking and placing objects from given continuous coordinates and propose an action attention module that predicts how each action will affect each object in the next state. They claim that by departing from the standard action-object dependence assumption made by previous works, the current model can effectively model the effect of future actions for structured tasks unseen during training. They finally propose to stack multiple GNN layers instead of a single layer as mostly practiced by earlier works. The authors evaluate their methodology on two pick-and-place tasks regarding building structures of abstract object shapes or cubes. The objective is to correctly predict the effect of certain action sequences in other blocks in order to build a certain structure (e.g. pile). They sample noise from the correct action sequence in order to generate negative examples for contrastive learning. They train in some sequence configurations and test for zero-shot performance in unseen tasks (new structures of objects). They report results for both mean square error of the block
position as well action ranking (how well the final state of the action sequence matches the final latent state).

**Summary Of The Review:**

This work presents an adaptation of an existing factored world model specifically tailored for robotic manipulation, using motion primitives (picking/placing objects) as the action space and pre-segmenting objects in order to factorize them into the state representation. The paper's novelty comprises the addition of an attention operation to contextualize future actions with latent object states. Even though experimental results suggest positively towards the potential of this method in task generalization, the lack of extensive comparison with previous factored
(structured) world models do not convince me 100% of the merits of the proposed methodology, especially when considering the capability of the standard C-SWM model to explore new object categories which are lost in the proposed model in favour of a "black-box" object detection system that factorizes the scene as input to the model.

---

> ### Author Response · Authors · 2021-11-20
> **Author Response**
>
> > Even though this work seems to be an extension of the factorized world model proposed by Kipf et al. (2020) there are no direct comparisons given with their model in the experiments section.
>
> This point was also raised by Reviewer WeZs. We plan to add this comparison to the revised version of our paper. The challenge here is that there is a mismatch in assumptions: we assume to have a factored state space, whereas the original C-SWM paper assumes to have a factored action space. We have designed heuristics to factor the action space (i.e. we check which objects have been affected by an action and provide the action only to their corresponding object slots) and we are now working on an experiment where we test the ability of the original C-SWM to learn a factored latent space based on the factored actions. We do not expect the model to do well, since learning state factorization is a very challenging task that often requires additional machinery.
>
> > The authors claim they wish to explore the potential of factored world models in generalizing to unseen tasks, but in the concerned environments, new "tasks" simply correspond to novel sequences of picking and placing actions (i.e. novel configurations), something previous works that impose permutation equivariance already demonstrate that they can achieve.
>
> Yes, here we define novel tasks as novel configurations of known objects. We would argue this is a reasonable definition; an example of this outside of our domain would be considering cooking a new recipe, which includes known ingredients, as a new task an agent should learn.
>
> Regarding previous work, various forms of generalization abilities of graph neural networks given by the property of permutation equivariance have definitely been demonstrated in many contexts. It is the main motivation for this paper. Our contribution here is that we demonstrate that graph neural networks can form a basis for learning generalizable forward models for robotic manipulation that accurately predict outcomes for sequences of actions that are long enough to accomplish a broad spectrum of related tasks, even for sequences of actions that were not seen during training. This is not something that has been demonstrated in previous work (at least, to the best of our knowledge).
>
> > This work's novelty focuses on the application of an attention mechanism and stacking multiple GNN layers. Both additions provide minor performance boosts (and not robustly across all experiments, as seen in rows 3, 4, and 5 of the bottom part of Table 1).
>
> This point was also raised by Reviewer WeZs and Reviewer iwuh. We added more experiments and analysis of attention in Section 3.3 and Figure 7. To summarize, we see a large benefit to using action attention when we train FWM with one GNN layer. We hypothesize that our full model, which uses four GNN layers, does not need action attention, since it has an abundance of representation capacity. We further show that with both one and four GNN layers, action attention learns interpretable weights (please see Figure 7). These weights can be useful for both model analysis and for downstream tasks.
>
> > A few grammatical mistakes and forgotten to actually cite works in the first two paragraphs of Section 2.
>
> We fixed the citations in the revision. Thank you for pointing that out.

---

> > ### Comment · Reviewer_4oxK · 2021-11-24
> > **Thank you for the clarifications.**
> >
> > Thank you for clarifying the concerns! It is an interesting read!

---

### Official Review · Reviewer_toZ5 · 2021-11-02

**Correctness:** 3
**Technical Novelty And Significance:** 2
**Empirical Novelty And Significance:** 2
**Recommendation:** 5
**Confidence:** 4

**Main Review:**

Strengths:

s1)  The paper proposes to leverage a graph neural network to extract the state feature and learn a latent dynamic model. The contrastive loss is applied to update the neural network weights.

s2) The paper is cleary-written and easy to follow.

Weaknesses:

w1) In Sec 2, "We will assume a setting in which the state of the world is represented as an image, which has been pre-processed into a factorized state s = $ <s_1 , s_2 , ..., s_K >$ in which each si is an image centered on the ith object". The assumption may not hold in many applications where some objects are fully occluded and can not be perceived.

w2) In the experiments, the evaluation setting is too simple. It would strengthen the paper if more complicated manipulation tasks (including more complex shapes and manipulation tasks pushing, pulling, throwing) are included.

**Summary Of The Paper:**

This work proposes a pipeline to learn factored world models to predict the robot actions’ effects. This work proposes to leverage a graph neural network to extract state information and adopt the contrastive loss to train the encoder and latent transition model. This paper applies the proposed approach to pick-and-place tasks with simple shapes, and extensive experiments indicate its effectiveness.

**Summary Of The Review:**

This work proposes a pipeline to learn factored world models to predict the robot actions' effects leveraging the graph neural network to encode the latent space and the contrastive loss to update the neural network's weight. However, the technical novelty is limited, and the current setting is too simple. It's unclear whether the proposed pipeline could work with complicated manipulation tasks.

---

> ### Author Response · Authors · 2021-11-20
> **Author Response**
>
> > w1) In Sec 2, "We will assume a setting in which the state of the world is represented as an image, which has been pre-processed into a factorized state $s = \langle s_1, s_2, ..., s_k \rangle$ in which each si is an image centered on the ith object". The assumption may not hold in many applications where some objects are fully occluded and can not be perceived.
>
> Occlusions do occasionally happen in our environment (e.g. a cube buried under a pile of other cubes). In these cases, the model gets a black image in the object slots corresponding to occluded objects, and it is trained to deal with these situations. It is true that our model relies on the Markov assumption; therefore, it does not have a memory of where an object was located before it got occluded. We believe it is reasonable to rely on the Markov assumption, as we can simply use as many RGB cameras (the ones we use for real-world experiments cost $25) as we need to observe all objects when working on a static robot. GNNs for POMDPs could be an interesting future direction.
>
> > w2) In the experiments, the evaluation setting is too simple. It would strengthen the paper if more complicated manipulation tasks (including more complex shapes and manipulation tasks pushing, pulling, throwing) are included.
>
> We added evaluation on a real-world dataset to the revised version of our paper (please see Section 3.4). We leave complex shapes and different motion primitives for follow-up work.

---

### Official Review · Reviewer_WeZs · 2021-11-03

**Correctness:** 4
**Technical Novelty And Significance:** 3
**Empirical Novelty And Significance:** 3
**Recommendation:** 6
**Confidence:** 4

**Main Review:**

The authors present an interesting approach with several novel ideas. Using an attention mechanism to address the complexity of environments containing larger numbers of objects to focus only those objects that are going to be affected by an action is interesting and novel. Based on the ablation study, however, that this mechanism does seem overly important. Would that change if the number of objects in the environment increases dramatically? The approach performs well in the scenario where the task was shown during training. More importantly, this is also the case for the scenario were the task that was not shown during training and shows the generalization abilities of the approach. The numbers for the two metrics (RMSE and Hits @ 1) are reasonable. The baselines, however, don't seem to be sufficient since (Qi et al. 2021) is either not failing to provide results or scores sufficiently worse compared to the presented approach, and the second baseline is a modified version of the presented approach. Veerapaneni et al. (2019) and (Kipf et al. 2020) use a similar test environment and (Huang et al., 2020) compare their results to (Kipf et al. 2020). The authors neither use those methods in their  evaluation nor do the discuss their reasons for excluding those. It would be great if the authors could provide more details on this part.
The submission is well written, the authors are able to communicate their ideas well, but placing the related work section at the end of the submission is confusing.

**Summary Of The Paper:**

The presented work addresses the task of robotic manipulation tailored to environments containing many objects. Task planning in such environments entails large number of possible combinations of actions given the number of objects. To address this challenge, the authors propose to use an attention module in combination with a graph neural network to predict the effects of the manipulation action, limiting the objects that need to be included into the execution planning. Experiments are conducted in simulated environment using a robotic arm interacting with two sets of shapes. The model performs well when for both test scenarios, one where the model operates on a task that is was trained on, and two on a task that was not presented during training showing the generalization capabilities of the presented approach.

**Summary Of The Review:**

The presented approach introduces interesting and novel ideas and shows that the model is able to generalize to tasks that it was not trained on. Only using one exterior baseline method that does not perform well seems not sufficient given that three related publications are evaluated using a similar setup. While the experimental section needs to be improved, I would rate the positive aspects the approach, generalization abilities and the novelty higher and would, therefore, argue for the submission to be accepted.

---

> ### Author Response · Authors · 2021-11-20
> **Author Response**
>
> > Based on the ablation study, however, that this mechanism does seem overly important. Would that change if the number of objects in the environment increases dramatically?
>
> We agree that we could see a larger benefit from the attention module if we increased the number of objects in the environment. In the current revision of our paper, we went in the opposite direction and added an experiment where we reduce model capacity by decreasing the number of GNN layers from four to one. With one GNN layer, the model has to rely on action attention and we see a large increase in performance (e.g. the block prediction error more than doubles for 1 GNN compared to 1 GNN + Attention). We also updated our visualizations to further illustrate the behavior of the attention module. Please see Table 1 (“FWM - 1 GNN” compared to “FWM - 1 GNN, No Att.”), Section 3.3 and Figure 7.
>
> > The baselines, however, don't seem to be sufficient since (Qi et al. 2021) is either not failing to provide results or scores sufficiently worse compared to the presented approach, and the second baseline is a modified version of the presented approach. Veerapaneni et al. (2019) and (Kipf et al. 2020) use a similar test environment and (Huang et al., 2020) compare their results to (Kipf et al. 2020). The authors neither use those methods in their evaluation nor do the discuss their reasons for excluding those. It would be great if the authors could provide more details on this part.
>
> Regarding comparing to (Kipf et al. 2020), which was also raised by  Reviewer toZ5:
>
> We plan to add this comparison to the revised version of our paper by the end of the rebuttal phase. The challenge here is that there is a mismatch in assumptions: we assume to have a factored state space, whereas the original C-SWM paper assumes to have a factored action space. We have designed heuristics to factor the action space (i.e. we check which objects have been affected by an action and provide the action only to their corresponding object slots) and we are now working on an experiment where we test the ability of the original C-SWM to learn a factored latent space based on the factored actions. We do not expect the model to do well, since learning state factorization is a very challenging task that often requires additional machinery.
>
> At present, we do not plan to compare to (Veerapaneni et al. 2019) and (Huang et al., 2020). The former demonstrate some robotic manipulation results (e.g. 3 block stacking with factored action space and 55% planning success rate), whereas the latter focus on toy domains and Atari games. Our aim is to further study the use of graph neural networks to model the physics of robotic manipulation separately from the task of unsupervised discovery of objects.

---

> > ### Comment · Reviewer_WeZs · 2021-11-30
> > **Rebuttal response**
> >
> > I would like to thank the authors for the very detailed rebuttal. Reading the other reviews has affirmed my concerns with respect to the evaluation of the presented work. While the authors where able to address some of my concerns with their great effort in conducting several new experiments, I do agree with the other reviewers in that a wider range of experiments, showcasing the benefits of the presented approach, would greatly strengthen the submission. I am lowering my rating to marginally above acceptance threshold and I would encourage the authors to continue their work.

---

### Author Response · Authors · 2021-11-20
**Paper Revision**

We thank the reviewers for their comments. We have uploaded a substantially revised version of our paper to address concerns about action attention and model evaluation. Sections, tables and figures that we added or changed are highlighted in blue.

Summary of changes:

1. We added results for action attention with one GNN layer (Table 1, FWM - 1 GNN, No Att.), which show a significant benefit to having action attention. We also extended visualizations of action attention weights (Figure 7). Note that using a single round of GNN message passing (i.e., one GNN layer) per time step is the setting used in prior work (Kipf et al., ICLR 2020).

2. We added an experiment with a real-world dataset in Section 3.4 and Table 2. We show that we can successfully use a model trained in simulation to make predictions about real-world manipulation.

3. We updated results for all models in Table 1 based on increased number of runs and bug fixes. In particular, we fixed a bug in the dataset collection for zero-shot generalization tasks in Cubes. In the current version, the gap between training and zero-shot generalization performance has decreased substantially.

We plan the do the following in the remainder of the rebuttal period:

1. We are working on a comparison to C-SWM, where we provide a heuristically factored action space to the model and test its ability to factor states and make predictions about the effects of actions. This experiment was suggested by Reviewers WeZs and toZ5.

2. We are considering adding an experiment, where action attention is used on a different domain than robotic manipulation, as suggested by Reviewer iwuh.

---

### Author Response · Authors · 2021-11-22
**Second Paper Revision**

In the second revision of our paper, we added a comparison to C-SWM, as suggested by Reviewers WeZs and toZ5. Please find C-SWM in Table 1, Appendix B.1 and Figure 9. Compared to FWM, C-SWM has to learn to factor the state space based on a factored action space. Even though we provided a heuristically factored action space (using privileged information), C-SWM fails to factor states in cube stacking into individual objects (Figure 9) and reaches performance similar to an unfactored baseline (Table 1, C-SWM vs FWM - No Factorization). This result indicates that it is non-trivial to use self-supervised object discovery (or state factorization) methods in our domains.

---

### Decision · Program_Chairs · 2022-01-20

**Decision:**

Reject

**Comment:**

This paper presents a GNN-based attention mechanism and tests it on a robotic stacking task.

While all the reviewers agree that this work is novel and interesting, they also are unanimous (even after the rebuttal) in pointing to the insufficient experimental evaluation of the proposed method.

I encourage the authors to incorporate the feedback of all the reviewers.